# Root Growth and Defense Response of Seedlings against *Fusarium oxysporum* in Sand Culture and In Vitro—A Comparison of Two Screening Approaches for Asparagus Cultivars

Roxana Djalali Farahani-Kofoet [ID], Jan Graefe and Rita Zrenner *[ID]

Leibniz Institute of Vegetable and Ornamental Crops (IGZ) e.V., 14979 Großbeeren, Germany; kofoetr@igzev.de (R.D.F.-K.); graefe@igzev.de (J.G.)
* Correspondence: zrenner@igzev.de; Tel.: +49-33701-78-216

**Abstract:** Two rapid asparagus (*Asparagus officinalis* L.) screening methods, in sand culture and in vitro, were tested to evaluate the response of young seedlings against *F. oxysporum* f. sp. *asparagi* (isolate Foa1). Root morphological parameters were evaluated and correlated with the symptomatology and expression of the defense-related genes at 5 and 7 dpi. In sand cultivation, the Foa1-inoculated cultivars showed no visible disease symptoms on their roots until 7 dpi. Two-factorial ANOVA statistics found no significant interaction between the cultivars and treatments for most root parameters but some differences between the cultivars. The in vitro Foa1-inoculated cultivars showed high susceptibility according to their symptomatology and differed greatly in the length of the primary root at 5 dpi. In some cultivars, the primary root length and root surface area were higher upon Foa1 inoculation. The expression changes were very different among the cultivars, with significant induction of *PR1*, *POX*, and *PAL* at 5 dpi in all cultivars in vitro but only in two cultivars in sand cultivation. The in vitro screening method, although more artificial, seemed to be more reliable than sand cultivation since the fungus was able to develop well in the culture medium. In sand-filled pots, the fungus may have been hindered in its development, even though a considerable higher amount of Foa1 was inoculated. In addition, the fungal growth was easily trackable in tubes, while in sand cultivation, the results were only visible after pulling the seedlings out of the pots 12 dpi.

**Keywords:** *Asparagus officinalis*; *Fusarium oxysporum* f. sp. *asparagi*; gene expression; root morphology; test tubes

## 1. Introduction

Fusarium wilt and rot of *Asparagus officinalis* L., caused by several *Fusarium* spp. such as *F. oxysporum* f. sp. *asparagi*, *F. proliferatum*, and *F. redolens*, have widely been considered as the most devastating disease in new and replanted asparagus fields [1,2]. Once the pathogen has been introduced into a field, its propagules (chlamydospores) can survive in the soil over a long period in the absence of the host plant, with the consequence being the loss of yields and acreage no longer suitable for further asparagus cultivation [3]. Avoiding fields where asparagus has previously been grown is the first solution for growers, but from a worldwide perspective, there is a lack of adequate agricultural land. Therefore, demand for effective control methods is inevitable. Over the past 75 years, disease management strategies to overcome the crown and root rot disease of asparagus have ranged from reducing the inoculum in soils and stress on the host to altering the soil environment to affect the disease [4]. Each of the suggested approaches such as chemical or biological control, incorporating organic soil matter, reducing stress in plants, changes in cultivation and mineral nutrition, as well as application of arbuscular mycorrhiza and biochar are to some extent promising and should be applied in a multifaceted way to achieve field longevity [4].

The integral part of the long-term effect of all these approaches is taking into account the resistance properties of the selected plant [4,5]. Breeding for resistance can be very difficult when no dominant gene is known and when new races of the pathogen develop and overcome host resistance [6]. As for tomato, there are varieties (tailored for greenhouse production) which are resistant to the common races of *F. oxysporum* f. sp. *lycopersici* based on a gene-for-gene relationship [6,7]. However, Fusarium-resistant *Asparagus officinalis* cultivars do not yet exist, but decades of breeders' attempts to breed robust and compact cultivars in combination with other desirable asparagus traits have resulted in a rich diversity of varieties. So far, only a few studies have been published demonstrating rapid testing methods to screen for Fusarium-resilient Asparagus genotypes. Stephens and Elmer (1988) [8] adopted and modified the in vitro assay introduced by Davis (1963) [9] for evaluating *Asparagus* spp. seedlings growing in test tubes in Hoagland's solution. They found differences in the disease reactions of various *Asparagus* spp. to *F. oxysporum* and rated the seedlings of *A. sprengeri* and *A. myersii* as resistant. Another test method was used by Kathe et al. (2019) [10], who assayed the susceptibility of various asparagus cultivars and wild asparagus species against *F. oxysporum* using digital image analysis. Through this non-destructive but artificial inoculation method, they found a resistant wild asparagus type and determined different levels of susceptibility for 16 asparagus genotypes to *F. oxysporum* as the first step for breeding Fusarium-resistant cultivars. Very recently, Jacobi et al. (2023) [11] introduced a screening test method on seedlings grown previously in sand culture, pulled out and inoculated by dipping the roots into *F. oxysporum* suspension. After incubation of the plants for 14 days on moistened filter paper, root reduction and root browning of the main root and lateral roots of the tested cultivars were scored using a phenotyping platform. Among the 17 screened cultivars and wild asparagus relatives, *A. aethiopicus* was the only symptomless variety combined with rapid accumulation of hydrogen peroxide in root cells, which is associated with high defense response. To date, it is not known how the growth characteristics of asparagus cultivars at the seedling stage are related to susceptibility and infection with *F. oxysporum* under differing growth conditions. A suitable seedling cultivation method in substrates would facilitate disease monitoring without injuring the root system. In addition, such a method would allow the investigation of as of yet missing molecular factors involved in the perception and subsequent activation of defense mechanisms in asparagus seedlings when challenged with *F. oxysporum*. The prerequisites for investigations on the molecular level are undamaged plants in good condition and assurance that infection with the pathogen has been successful, which can be achieved less well with the screening methods mentioned above. Linking the resilience of asparagus to *F. oxysporum* with early manifested root growth traits would help to find tolerant varieties within a short period of time.

In this work, we investigate the feasibility of the two test methods for continuous monitoring of disease progression in relation to symptomatology, root morphology, and defense-specific gene expression upon infection with *F. oxysporum* f. sp. *asparagi*. To this end, we contrast two screening methods, one in vitro and one in sand culture, while testing the selected asparagus cultivars at the seedling stage and discussing the advantages and disadvantages.

## 2. Materials and Methods

### 2.1. Asparagus Cultivars and Seed Surface Disinfection for the In Vitro and Pot Trials

For both pathogen–host interaction test methods, the asparagus cultivars 'Backlim', 'Gijnlim', 'Grolim', 'Vitalim', 'Xenolim' (Limgroup; NV Horst, the Netherlands), 'Rapsody', 'Ramires' (Südwestsaat; Rastatt, Germany), and additionally 'Fortems' (Nunhems; Marbach am Neckar, Germany) for sand culture were assayed. For seed surface disinfection, the seeds were soaked for 30 minutes (min) in warm water (50 °C), washed in sterile deionized water plus Tween 20 (0.1%) (Merck KGaA; Darmstadt, Germany) for 5 min, then soaked in ethanol (70%) plus Tween 20 (0.1%) for 5 min, sterilized in $NaClO_3$ (12%) for 10 min, and washed (4×) with sterile water.

### 2.2. Pathogen Isolate F. oxysporum f. sp. asparagi

The *F. oxysporum* f. sp. *asparagi* single spore isolate Foa1 tested as aggressive [12] was used for inoculation of the asparagus roots in both cultivation systems. Maintenance of the pathogen took place on potato dextrose agar (PDA; Merck, Germany) at 23 °C and in darkness. Spore suspensions with the respective conidia concentration of $3 \times 10^6$ conidia $mL^{-1}$ used in both experimental systems were prepared as previously described [12].

### 2.3. Sand Culture Experiment—Cultivation in Pots and Pathogen Inoculation

Four disinfected seeds of the respective cultivars were seeded in pots (12.8 cm × 12.8 cm × 20 cm; 2 L) filled with quartz sand as substrate. To avoid desiccation during germination, the pots were watered with tap water and covered with transparent film until the first shoots emerged. The cultivation of the plants took place in growth chambers at 25/20 °C, 85/75% rel. humidity, and a day/night photoperiod of 12/12 h at 400 µmol $m^{-2}$ $s^{-1}$ light, except for the first week as asparagus is a dark germinator. The temperature conditions were reduced to 20/15 °C after two weeks. Twice a week, nutrient solution was poured into trays, with the pots placed in the trays [12]. When the first shoots emerged, 14 days after seeding, each plant was inoculated with 100 µL of Foa1 with $3 \times 10^6$ conidia $mL^{-1}$ or with sterile deionized water as the control. To complete this, the substrate was very carefully removed from the top three centimetres of the root and the respective solution was pipetted directly onto the root, which was immediately covered again with quartz sand.

Three pots with four seedlings each with the same cultivar and inoculum concentration for three assessment times represented one block with randomized setting. In total, three blocks were included in the experimental design (n = 3).

### 2.4. Disease Assessment, Imaging Analysis of Roots, Plant Sampling, and Molecular Analysis

The plants were removed from the substrate and assessed visually for disease and growth development at 5, 7, and 12 days post inoculation (dpi). Here, the first three thick roots were defined as the primary roots, and the fibrous roots originating from the primary roots were defined as the laterals. Plant growth was recorded at 5 dpi by counting the number of stems and primary and lateral roots and by measuring the length of the first primary root with a ruler.

For the root morphological analysis, the roots were washed and scanned at 600 dots per inch. The segmented and skeletonized roots where analyzed for their total root length as described by Kimura et al. (1999) [13], whereas their root surface area was derived from the projected root area multiplied by π, i.e., assuming a cylindrical shape. The number of lateral roots was estimated from a marked preliminary list of root tips [14] which was further validated and reduced judged on an observable local increase in tip diameter.

Root sampling for gene expression analyses was performed at 5 dpi and was performed by removing $3 \times 3$ plants per cultivar and treatment. The Foa1- and water-inoculated plants were carefully pulled out of the sand substrate and cleaned of any remaining sand. Shoots were cut from the roots with a sterilized scalpel, and their fresh weight was recorded separately (Supplementary Table S1). The total root system and a 2 cm piece of the shoot base were frozen at −80 °C for molecular analyses. The frozen plant material was ground in an orbital ball mill twice for one min at 30 Hz $s^{-1}$ (MM400; Retsch GmbH; Düsseldorf, Germany) with two balls of 4 mm-diameter stainless steel. RNA was extracted from 100 mg of ground tissue using an innuPREP Plant RNA Kit (Analytik Jena; Jena, Germany), quantified spectrophotometrically at 260 nm (NanoPhotometer NP80; Implen GmbH; Munich, Germany), and quality controlled using a 2100 Bioanalyzer and an RNA 6000 Nano kit (Agilent Technologies; Santa Clara, CA, USA). Single-stranded cDNA synthesis was performed with 1 µg total RNA using an iScript cDNA Synthesis Kit (Bio-Rad Laboratories GmbH; Feldkirchen, Germany) in a 25 µL reaction and diluted 10-fold. RT-qPCR was performed in 96-well reaction plates on a Thermal Cycler CFX96 C1000 Touch (Bio-Rad Laboratories GmbH) with a thermal profile of 95 °C for 5 min,

40 cycles of 95 °C for 15 s, and 60 °C for 1 min, followed by dsDNA melting curve analysis. Each reaction was performed in 10 µL with 200 nM of each primer, 3 µL of cDNA (1:10), and 5 µL of SensiFAST SYBR No-ROX Kit (BioCat GmbH; Heidelberg, Germany). Data were collected and compiled using CFX Manager Software 3.0 (Bio-Rad Laboratories GmbH). Relative transcript levels were normalised on the basis of the expression of the invariant control elongation factor 1-alpha (XM_020411981), calculating $\Delta$Cq as the difference between the control and target products ($\Delta$Cq = Cq$_{gene}$ − Cq$_{EF1}$). Differences in relative expression levels between the treated samples were calculated as −$\Delta\Delta$Cq = $\Delta$Cq (*Fusarium* spp.-inoculated sample) − $\Delta$Cq (mock inoculated sample). Oligonucleotide primers: elongation factor 1-alpha (XM_020411981, *EF1*), Ao-RT-EF1alfa2f: TTGATAGGC-GATCGGGTAAG, Ao-RT-EF1alfa2r: CTCATGTCCCTCACAGCAAA; pathogenesis-related protein 1 (XP_020276576, *PR1*), Ao-RT-PR1f2: TGTTCGAATCTGCCACTACT, Ao-RT-PR1r: TGCCTTCATGTGGTTGGTTA; cationic peroxidase 1-like (XM_020420634, *POX*), Ao-RT-POX2f1: GCTTCAGCCCAGTTATCGTC, Ao-RT-POX2r1: CATTGACGAAGCAATCATGG; phenylalanine ammonia-lyase (XM_020404206, *PAL*), Ao-RT-PALf: GTAAACGACAACC-CGCTCAT, and Ao-RT-PALr: AGCTCCGATACCTGAGCAAA.

*2.5. In Vitro Experiment—Cultivation in Sterile Tubes and Pathogen Inoculation*

Seed germination (60 seeds per cultivar) took place on sterile filter paper discs in Petri dishes (Ø 15 cm) moistened with sterile water and sealed with Leucopor (BSN medical GmbH; Hamburg, Germany) in an incubator at 23 °C and in darkness. If necessary, sterile water was added with a pipette to the filter paper under sterile conditions. Nine days after sowing, the germinated asparagus seeds were transferred to individual sterile tubes (De Wit Plastic; Duchefa; RV Haarlem, the Netherlands) containing growth medium (20 mL; 0.3% gelrite (Duchefa), 2.45 g MS nutrient solution + vitamins, and MES-buffer (Duchefa), 1 L deionized water, pH 5.8) and placed on polystyrene plates (Duchefa). Cultivation of the seedlings took place in growth chambers at 25/20 °C, 85/75% rel. humidity, and a day/night photoperiod of 12/12 h at 400 µmol m$^{-2}$ s$^{-1}$ light. The seedlings had developed a 3 cm-long primary root within 7 days and were prepared for inoculation with the pathogen.

The root system of the 16-day-old asparagus was inoculated with a 5 µL spore suspension ($3 \times 10^6$ conidia mL$^{-1}$) by inserting the pipette (1–2 cm depth) into the culture medium close to the root. The sealed culture tubes were placed in the growth chamber again for further growth. The control plants were inoculated with sterile deionized water. For the control of fungus growth, 10 tubes containing the nutrient medium without plants were also inoculated as above.

The in vitro experiment was performed twice with 4 blocks including 5 plants/tubes each (20 plants per cultivar) (n = 4). The tubes (9 cultivars × 2 treatments × 20 plants) were distributed in a randomized block design.

*2.6. Plant Development, Disease Assessment, Imaging Analysis of Roots, and Molecular Analysis*

Plant growth, disease progression in the roots and aboveground parts (shoots and phylloclades), and fungal growth were evaluated visually at 3 and 5 dpi in the tubes and at 7 dpi after pulling the plants out of the medium. Plant growth was recorded at 5 dpi by counting the number of shoots and primary and lateral roots and by measuring the length of the first primary root with a ruler while the plants were still in the sterile tubes. Fungal length growth was likewise measured alongside the 1st primary root with a ruler at 5 dpi. Symptoms on the shoots were assessed by recording visible changes on the shoots and foliage ranging from chlorotic to necrotic discolouration to wilting and dying-off. Disease symptoms on the roots were rated by counting the lesions (light brown, glassy, or brown coloration) on the primary and lateral roots. The percentage of symptomatic primary and lateral roots was calculated by relating them to those without symptoms.

After visual assessment, the roots were scanned at 1200 dots per inch and analyzed as described above.

For the gene expression analyses at 5 dpi, the Foa1- and water-inoculated plants were smoothly pulled out of the nutrition medium. Shoots were cut from the roots with a sterilized scalpel, and their fresh weight was recorded separately (Supplementary Table S2). Sample processing and the molecular analyses were performed as described above.

*2.7. Statistical Analysis and Experimental Design*

The two-factorial experiments were evaluated for possible treatment, cultivar, and interaction effects using ANOVA in Statistica®v13.5 (TIBCO Software Inc.; Palo Alto, CA, USA). For comparison of fungal growth in the tubes, one-factorial ANOVA was performed. Significant differences between the cultivars and pairwise significant differences between the treatments were determined by Tukey's test ($p < 0.05$). Pairwise significant differences between the molecular results were calculated by a *t*-test.

**3. Results**

*3.1. Sand Culture Experiment*

In this experiment, eight cultivars were seeded in sand-filled pots and inoculated with 100 µL of Foa1. After careful extraction of the plants, their root morphology and any disease symptoms in the roots were evaluated, and gene expression analyses were performed.

3.1.1. Morphological Assessment

The visually assessed roots did not show any symptomatic lesions at 5 dpi in sand culture. The roots assessed at 7 dpi were slightly yellow–brown coloured, particularly the primary root. At the end of the experiment at 12 dpi, all Foa1-inoculated roots showed brown tips and scattered brownish lesions. Statistical analyses of the root and shoot parameters showed significance among the cultivars in all traits, but there was no significant interaction between the cultivar and treatment for any of the measured traits (Table 1).

**Table 1.** Results of the variance analysis, *p*-values for treatment, cultivar, and interaction effects on the fresh masses (FM) of the roots and shoots, the 1st primary root length, the number of primary roots (n), the number of lateral roots (n), the total root length, and the root surface area in sand culture.

| Sand Culture | Root FM | Shoot FM | 1st Primary Root Length | Primary Roots (n) | Lateral Roots (n) | Total Root Length | Root Surface Area |
|---|---|---|---|---|---|---|---|
| Cultivar | 0.000 * | 0.006 * | 0.012 * | 0.008 * | 0.002 * | 0.000 * | 0.000 * |
| Treatment | 0.596 | 0.980 | 0.687 | 0.134 | 0.005 * | 0.034 * | 0.030 * |
| Cultivar × treatment | 0.407 | 0.656 | 0.821 | 0.350 | 0.156 | 0.431 | 0.361 |

* Significant at $p < 0.05$.

The mean number of primary roots and the length of the 1st primary root were assessed at 5 dpi. Among the cultivars, 'Vitalim' had the longest 1st primary root, differing significantly only from 'Backlim'. Except for 'Gijnlim', there was a significant difference between the water and Foa1 inoculation of each cultivar (Figure 1a). 'Backlim' and 'Ramires' were the slowest cultivars in primary root formation (Figure 1b). When looking at the lateral roots, 'Ramires' and 'Xenolim' had significantly fewer lateral roots than 'Vitalim' and 'Rapsody' (Figure 1c). 'Vitalim', 'Rapsody', 'Gijnlim', and 'Backlim' showed significantly higher total root lengths (Figure 2a), while 'Vitalim', 'Rapsody', and 'Gijnlim' also stood out with significant higher root surface areas (Figure 2b).

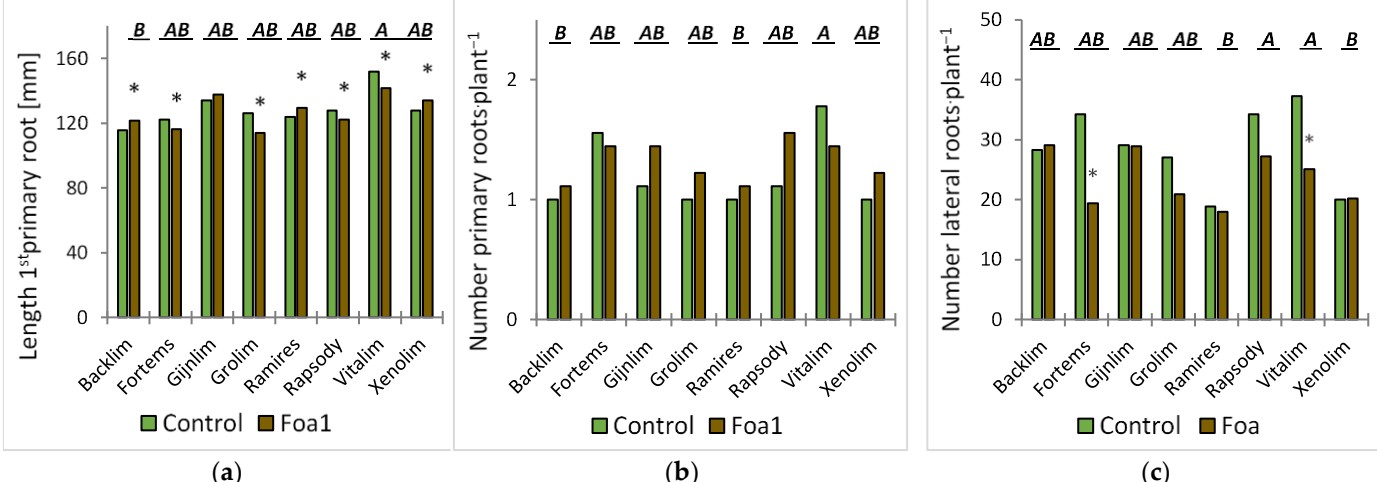

**Figure 1.** Length of the first primary root (mm) (**a**), number of primary roots (**b**), and number of lateral roots (**c**) of 'Backlim', 'Fortems', 'Gijnlim', 'Grolim', 'Ramires', 'Rapsody', 'Vitalim', and 'Xenolim' in sand culture at 5 dpi. The plants were inoculated with *Fusarium oxysporum* f. sp. *asparagi* isolate Foa1 with $3 \times 10^6$ conidia mL$^{-1}$ or water (control) (n = 3). Different bold italic letters indicate significance among the cultivars. Asterisks stand for significance between the control and the respective Foa1-inoculated cultivar (Tukey's test, $p < 0.05$). HSD$_{\text{length primary roots}}$ = 2.51; HSD$_{\text{number primary roots}}$ = 0.53; HSD$_{\text{lateral roots}}$ = 10.81.

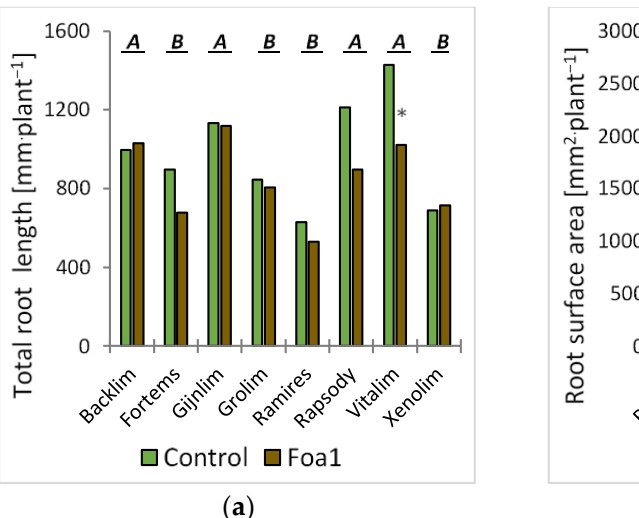
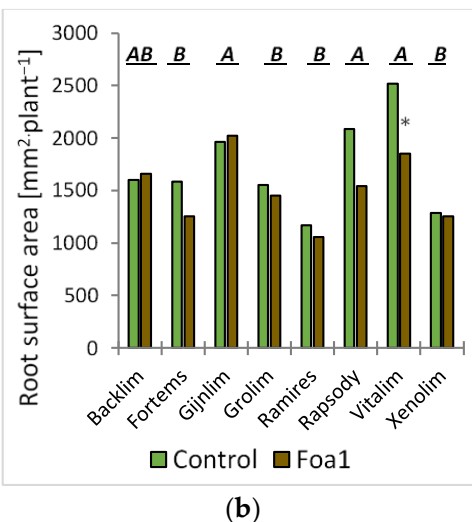

**Figure 2.** Total length (mm) (**a**) and surface area (mm$^2$) of the primary and lateral roots (**b**) of 'Backlim', 'Fortems', 'Gijnlim', 'Grolim', 'Ramires', 'Rapsody', 'Vitalim', and 'Xenolim' in sand culture at 7 dpi. The plants were inoculated with *Fusarium oxysporum* f. sp. *asparagi* isolate Foa1 with $3 \times 10^6$ conidia mL$^{-1}$ or water (control) (n = 3). Different bold italic letters indicate significance among the cultivars. Asterisks denote significance between the control and the respective Foa1-inoculated cultivar (Tukey's test, $p < 0.05$). HSD$_{\text{root surface area}}$ = 597.75; HSD$_{\text{total roots length}}$ = 380.95.

### 3.1.2. Gene Expression

Infection-related gene expression differences in *PR1*, *POX*, and *PAL* in the roots were determined. These genes represent different aspects of the defense strategies and revealed differences between the cultivars. Significant induction of *PR1* was detected in 'Rapsody', 'Xenolim', 'Vitalim', and 'Grolim', while 'Fortems' and 'Backlim' showed reduced expression of *PR1* at 5 dpi (Figure 3). The *POX* expression differences were somehow similar, showing significant induction in 'Rapsody' and 'Xenolim' and repression in 'Fortems' and 'Gijnlim'. *PAL* showed the lowest and least significant expression differences at 5 dpi, but

with comparable changes, i.e., induction in 'Rapsody' and 'Xenolim' and repression in 'Backlim', 'Fortems', and 'Gijnlim'.

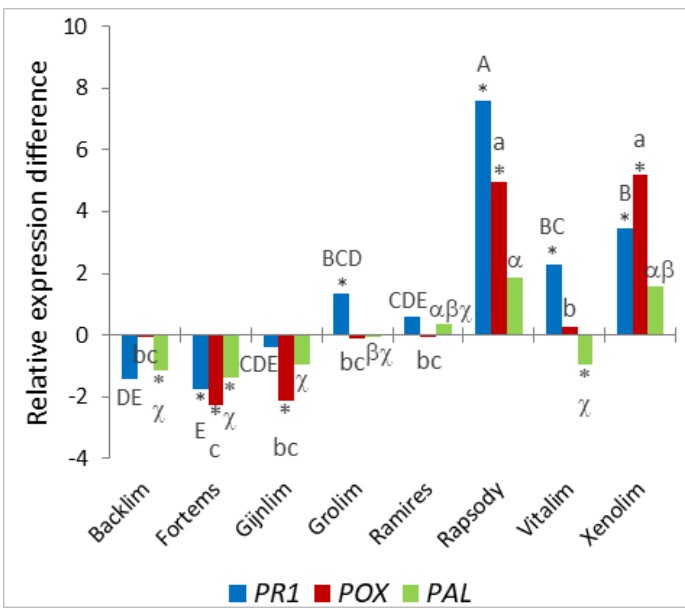

**Figure 3.** Relative expression difference in *PR1*, *POX*, and *PAL* measured in the roots of the asparagus cultivars 'Backlim', 'Fortems', 'Gijnlim', 'Grolim', 'Ramires', 'Rapsody', 'Vitalim', and 'Xenolim' grown in sand culture and inoculated with 100 μL *Fusarium oxysporum* f. sp. *asparagi* isolate Foa1 with $3 \times 10^6$ conidia mL$^{-1}$ at 5 dpi (n = 3). Different letters indicate significance among the cultivars within each gene expression analysis ($p < 0.05$). Asterisks denote significance between the control and the respective Foa1-inoculated cultivar (*t*-test, $p < 0.05$).

### 3.2. In Vitro Experiment

The same set of asparagus cultivars except 'Fortems' were cultivated in vitro in nutrient medium, inoculated with the Foa1 isolate, and cultivated aseptically in climate chambers to track root, disease, and fungus development in the culture tubes (Figure 4).

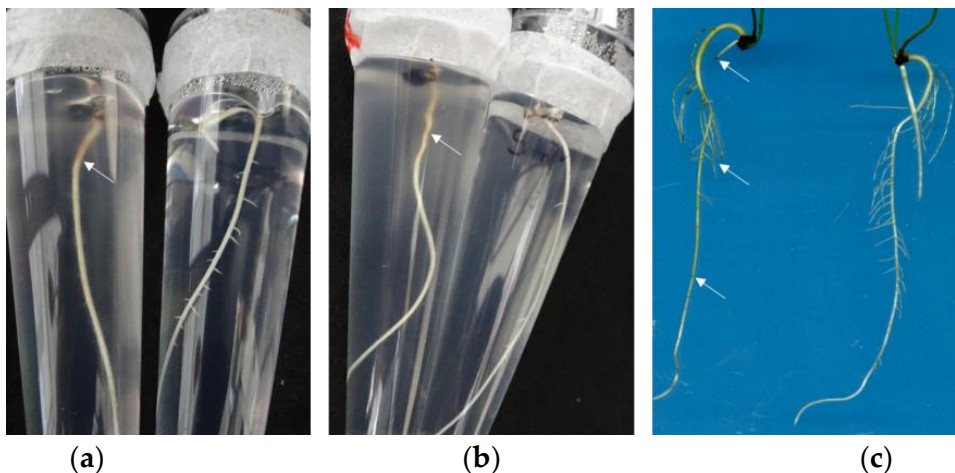

(**a**)            (**b**)            (**c**)

**Figure 4.** Images of in vitro culture grown plant roots of the cultivars 'Backlim' (**a**), and 'Ramires' (**b**) at 5 dpi and 'Gijnlim' (**c**) at 7 dpi after pulling out the plants for root imaging and disease assessment. The plants were either inoculated with *Fusarium oxysporum* f. sp. *asparagi* isolate Foa1 (**left**) or water (**right**). Arrows point to brown lesions on the first primary roots (**a,b**) and on the brown root tip of the primary root and the lateral roots (**c**).

3.2.1. Morphological Assessment

During disease development, the first symptoms caused by Foa1 were observed with the appearance of small lesions at 5 dpi on the roots. Fine lesions were detected on the primary roots at the junctions with the lateral roots. The lateral roots also showed small brown coloration of the tip. The discolorations and lesions expanded to 7 dpi, with occasional translucent lesions on the roots indicating disintegrated cells within the affected region (Figure 4). The lowest percentage of symptomatic primary and lateral roots at 7 dpi was found for 'Xenolim', differing significantly from 'Backlim' and 'Gijnlim' (Figure 5). Throughout the experiment, which ended at 7 dpi, the shoots did not show any symptoms of disease.

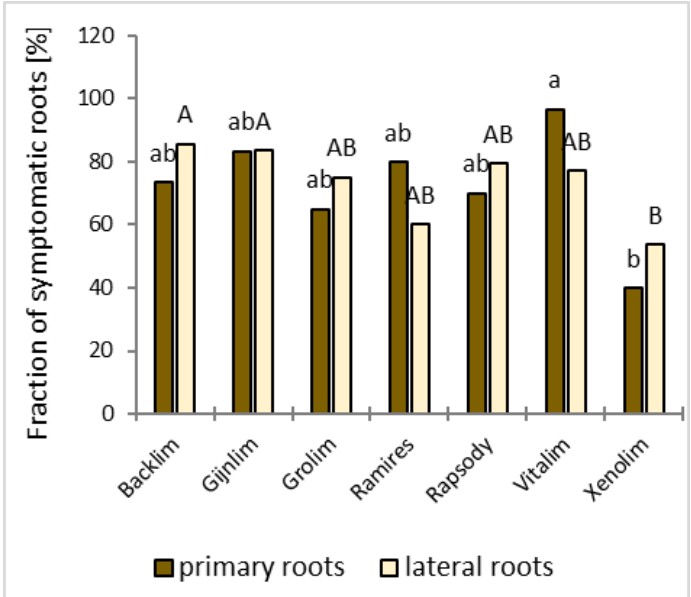

**Figure 5.** Percentage of symptomatic primary and lateral roots of the asparagus cultivars 'Backlim', 'Gijnlim', 'Grolim', 'Ramires', 'Rapsody', 'Vitalim', and 'Xenolim' at 7 dpi. The plants in in vitro culture were inoculated with *Fusarium oxysporum* f. sp. *asparagi* isolate Foa1 or water (n = 4). Different letters indicate significance among the cultivars' primary roots (lower case) and the cultivars' lateral roots (upper case) (Tukey's test, $p < 0.05$). HSD$_{symptoms primary roots}$ = 65.36; HSD$_{symptoms lateral roots}$ = 30.25.

Statistical analyses of the root and shoot parameters showed significance among the cultivars in all traits (Table 2). Significance in the interaction between the cultivars and treatments was only found for the primary root length (Figure 6a), shoot numbers (Figure 6d), and root surface area (Figure 7b). A significant difference in treatment was determined only in the primary root length (Figure 6a), the number of lateral roots (Figure 6c), and the shoot numbers (Figure 6d). Within the analyzed growth period until 5 dpi 'Vitalim', 'Backlim', and 'Gijnlim' had the longest 1st primary roots, but their growth was not affected by inoculation with Foa1 (Figure 6a). In contrast, the 1st primary roots of 'Grolim', 'Ramires', and 'Rapsody' were significantly the shortest, and the Foa1-inoculated plants grew significantly longer than their respective controls, also observed for 'Xenolim' (Figure 6a). When looking at primary root formation, there was no significant difference among the water- and Foa1-inoculated cultivars, except for 'Grolim' (Figure 6b). 'Vitalim' rapidly developed the next primary root followed by 'Gijnlim'. 'Rapsody', 'Ramires', 'Xenolim', and 'Grolim', while 'Backlim' developed until 5 dpi the shortest primary roots (Figure 6b). The number of lateral roots was significantly the highest in 'Vitalim' and 'Gijnlim', followed by 'Backlim' and 'Xenolim', while Grolim', 'Rapsody', and 'Ramires' developed the shortest lateral roots (Figure 6c). Among the cultivars 'Gijnlim', 'Backlim', and 'Vitalim' produced

the most shoots until 5 dpi. Except for 'Gijnlim' and 'Grolim', Foa1 treatment reduced shoot formation but not significantly (Figure 6d).

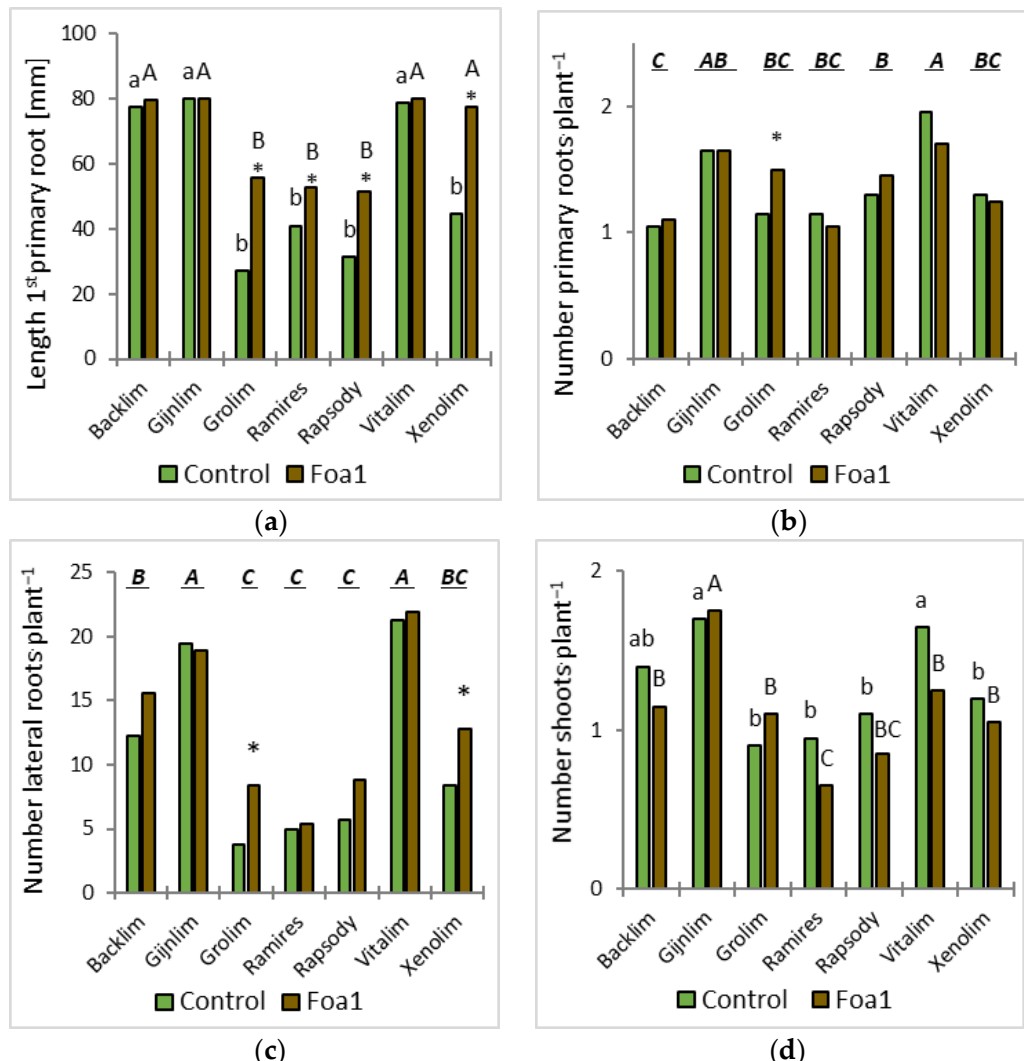

**Figure 6.** The length of the first primary root (mm) (**a**), number of primary roots (**b**), number of lateral roots (**c**), and number of shoots (**d**) of 'Backlim', 'Gijnlim', 'Grolim', 'Ramires', 'Rapsody', 'Vitalim', and 'Xenolim' at 5 dpi. The in vitro culture plants were inoculated with *Fusarium oxysporum* f. sp. *asparagi* isolate Foa1 ($3 \times 10^6$ conidia mL$^{-1}$) or water (control) (n = 4). Different letters indicate significance among the water-inoculated (lower case) and Foa1-inoculated (upper case) cultivars. Different bold upper-case letters in italics indicate significance among the cultivars. Asterisks denote significance between the control and the respective Foa1-inoculated cultivars (Tukey-test, $p < 0.05$). HSD$_{\text{length primary.roots}}$ = 2.35; HSD$_{\text{number primary.roots}}$ = 0.28; HSD$_{\text{lateral roots}}$ = 4.44; HSD$_{\text{shoots}}$ = 0.41.

**Table 2.** Results of the variance analysis, *p*-values for treatment, cultivar, and interaction effects on the fresh masses (FM) of the roots and shoots, the 1st primary root length, the number of primary roots (n) and shoots (n), the number of lateral roots (n), the total root length, the root surface area, and the Foa1 growth in the in vitro culture.

| In Vitro Culture | Root FM | Shoot FM | 1st Primary Root Length | Primary Roots (n) | Lateral Roots (n) | Shoots (n) | Total Root Length | Root Surface Area | Foa1 Growth |
|---|---|---|---|---|---|---|---|---|---|
| Cultivar | 0.000 * | 0.010 * | 0.000 * | 0.000 * | 0.000 * | 0.000 * | 0.000 * | 0.000 * | |
| Treatment | 0.376 | 0.176 | 0.000 * | 0.671 | 0.004 * | 0.002 * | 0.196 | 0.890 | 0.000 * |
| Cultivar × treatment | 0.315 | 0.175 | 0.007 * | 0.59 | 0.441 | 0.027 * | 0.078 | 0.014 * | |

* Significant at $p \leq 0.05$.

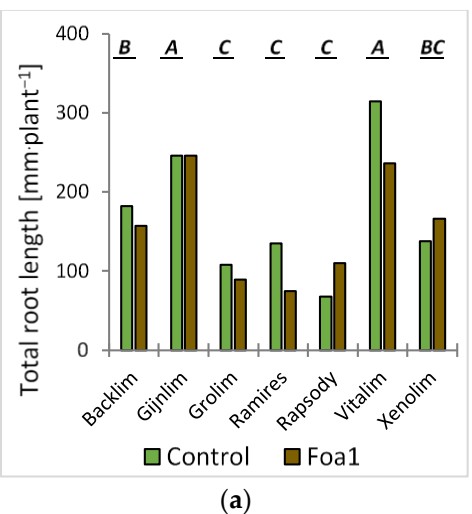

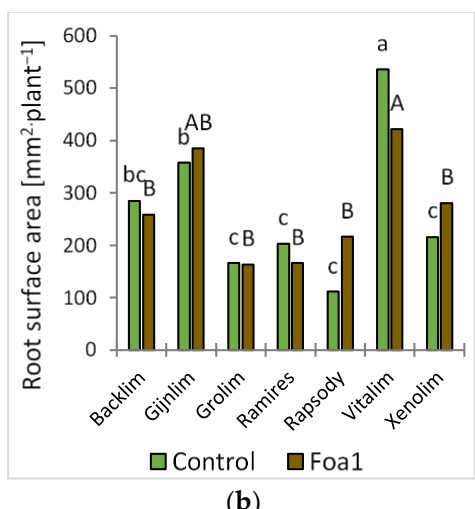

(**a**)　　　　　　　　　　　　　　　　　　　　　　　　(**b**)

**Figure 7.** Total length (mm) (**a**) and surface area (mm²) of the primary and lateral roots (**b**) of 'Backlim', 'Gijnlim', 'Grolim', 'Ramires', 'Rapsody', 'Vitalim', and 'Xenolim' at 7 dpi. The in vitro culture grown plants were inoculated with *Fusarium oxysporum* f. sp. *asparagi* isolate Foa1 ($3 \times 10^6$ conidia mL$^{-1}$) or water (control) (n = 4). Different letters indicate significance among the water-inoculated (lower case) and Foa1-inoculated (upper-case letters) cultivars. Different bold upper-case letters in italics indicate significance among the cultivars (Tukey's test, $p < 0.05$). HSD$_{\text{total roots length}}$ = 113.01; HSD$_{\text{roots surface area}}$ = 133.44.

The total root length at 7 dpi was highest for 'Vitalim' and 'Gijnlim' and lowest for 'Grolim', 'Ramires', and 'Rapsody'. Affection by Foa1 varied among the cultivars and did not differ significantly from water inoculation of the respective cultivars (Figure 7a). There was interaction between root surface area and inoculation. The root surface area of Foa1-inoculated 'Vitalim' differed greatly from that of 'Backlim', 'Grolim', 'Ramires', 'Rapsody', and 'Xenolim' (Figure 7b). For the two latter traits, no significant difference between the control and the Foa1-inoculated plants of each cultivar was recorded (Figure 7). Simultaneously, the vertically downwards growth of Foa1 along the roots within the tubes was measured (Figure 4). The growth of the fungus correlated with the length of the 1st primary root. Foa1 mycelium grew best with 'Backlim', 'Gijnlim', and 'Vitalim' followed by 'Grolim', 'Ramires', and 'Xenolim'. The lowest fungus growth was significant in the tubes with 'Rapsody' and in the control tubes without any asparagus plant roots (Figure 8).

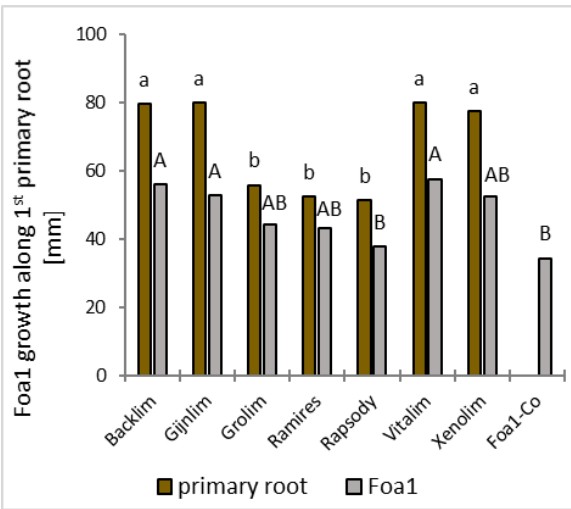

**Figure 8.** Growth of the *Fusarium oxysporum* f. sp. *asparagi* isolate Foa1 along the first primary root of the infected asparagus cultivars 'Backlim', 'Gijnlim', 'Grolim', 'Ramires', 'Rapsody', 'Vitalim', and 'Xenolim' and without the host plant (Foa1-Co) at 5 dpi (n = 4). Different letters indicate significance among the cultivars (lower case) or Foa1 growth (upper case) (Tukey's test, $p < 0.05$). $HSD_{Foa1\ growth} = 1.48$.

### 3.2.2. Gene Expression

Among the seven cultivars grown in vitro in test tubes, *PR1*, *POX*, and *PAL* were analyzed. All Foa1-inoculated varieties showed significant induction of these three genes in the asparagus roots at 5 dpi (Figure 9a). Induction of *PR1* was lowest in 'Xenolim' and 'Rapsody' and highest in 'Backlim'. *POX* induction was highest in 'Xenolim' and 'Vitalim', and *PAL* showed a significant and the highest increase in the 'Vitalim' and 'Gijnlim' roots (Figure 9a). *PR1* was also significantly induced in the shoots of all cultivars, with the highest expression increase in 'Grolim', although the most significant and strongest induction of *POX* was present only in 'Vitalim', and no significant change in *PAL* expression was detected in any of the cultivars' shoots (Figure 9b).

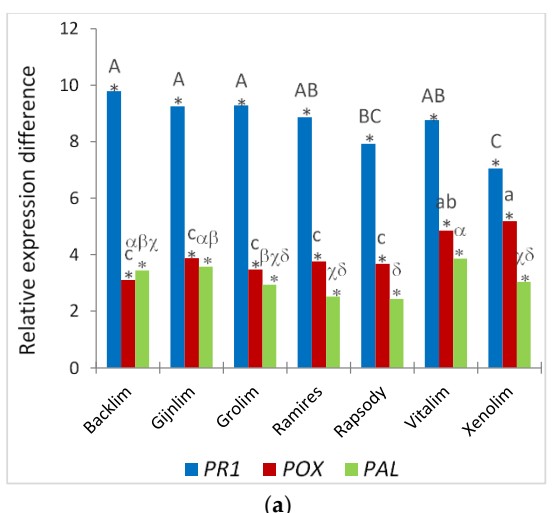

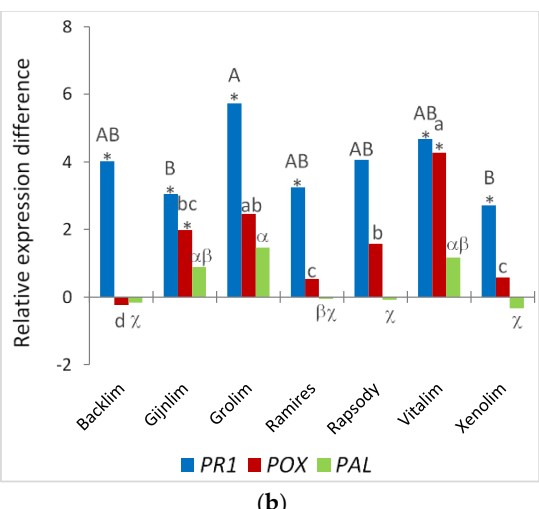

(**a**)　　　　　　　　　　　　　　　　(**b**)

**Figure 9.** Relative expression difference in *PR1*, *POX*, and *PAL* measured in the roots (**a**) and shoots (**b**) of the asparagus cultivars 'Backlim', 'Gijnlim', 'Grolim', 'Ramires', 'Rapsody', 'Vitalim', and 'Xenolim' grown in vitro and inoculated with *Fusarium oxysporum* f. sp. *asparagi* isolate Foa1 at 5 dpi (n = 5). Different letters indicate significance among the cultivars within each gene expression analysis ($p < 0.05$). Asterisks denote significance between the control and the respective Foa1-inoculated cultivar (*t*-test, $p < 0.05$).

## 4. Discussion

In this work, we hypothesized that infection of asparagus plants at the seedling stage with *F. oxysporum* would affect root development, root growth, and defense response differently depending on the cultivar. We investigated the reaction of 16-day-old plants of eight selected asparagus cultivars to *F. oxysporum* f. sp. *asparagi*, isolate Foa1 in pots filled with sand. The cultivation conditions were as close as possible to those in a greenhouse, except for the use of pure sand as the growing medium. Sand as substrate was used to allow for careful removal of the plant roots without damaging the primary and lateral roots, essential for analyses of root morphology and gene expression. In our sand cultivation test, all cultivars were susceptible to Foa1 at 12 dpi. Disease symptoms on the roots were not clearly visible until 7 dpi. This is probably due to the fact that Foa1 grows near the roots, whereas the sandy environment is not very suitable for fungal growth. Our results are in agreement with the results of Jacobi et al. (2023) [11], who detected brown lesions on Foa1-inoculated roots of all asparagus cultivars 14 dpi by a phenotyping platform, albeit their screening method was under more artificial incubation conditions. They initially grew asparagus genotypes in sand culture but pulled out the seedlings for drip-inoculation with *F. oxysporum* and incubated the inoculated plants for 14 days on moistened filter paper, i.e., outside any substrate.

We observed some significant root morphological differences between Foa1- and the respective water-inoculated plants in sand culture. For example, the Foa1-inoculated 'Backlim' and 'Xenolim' showed increased primary root length with an induction of newly formed primary roots, while the number of lateral roots, total root length, and root surface area remained unaffected. In contrast, the Foa1-inoculated 'Vitalim' showed a shorter primary root length with a reduction in newly formed primary roots and a significant reduction in the number of lateral roots, total root length, and root surface area, while 'Gijnlim' did not show significant alteration in any of these measured traits. We consider this result as a varietal response that should be further investigated in more detail in future trials, especially with regard to a possible link with susceptibility. This is particularly relevant in light of the fact that root morphological characteristics are more consistent indicators of yield loss than root rot severity [15] and that a significant increase in both specific root length and surface area occurs in response to pathogen invasion, regardless of pathogen density [16]. In contrast, Jacobi et al. (2023) [11] could not detect a significant reduction in root length in some tested genotypes classified as susceptible to Foa1, which is why they exclude root length reduction as a suitable selection criterion for Foa1 resistance.

In addition to root morphological traits as indicators of susceptibility, specific gene expression differences have long been studied as markers of infection. In this study, we analyzed the differential expression of pathogenesis-related 1 gene (*PR1*), a gene encoding a cell-wall-bound cationic peroxidase (*POX*), and a gene that encodes phenylalanine ammonia lyase (*PAL*), which represent different response strategies upon pathogen attack. *PR1* gene product is induced by pathogen attack via the salicylic acid (SA) pathway and may have an antimicrobial function [17]. The *POX* gene product may play a role in cell wall lignification [18], thus making it more difficult for pathogens to invade. Phenylalanine ammonia lyase (PAL) is the key enzyme of phenylpropanoid biosynthesis. These metabolites have many different functions in defense, and PAL itself contributes to SA biosynthesis [19,20]. Gene expression analyses of *PR1*, *POX,* and *PAL* showed some differences in the induction of these defense-related genes between cultivars; however, these differences could not be related to morphological traits. We suggest that this result also points to variety-specific differences in the very early response of the plant to pathogen attack, which should be investigated in the future, especially with regard to a possible connection with susceptibility in a temporal and inoculum-dependent manner.

Since we could not directly follow the growth of the pathogen in pots filled with sand, we carried out an in vitro test, in which we observed the development and growth of the roots at three time points, while simultaneously tracking the growth of the fungus downwards along the roots in the tubes. Through this method, we could prove that the

pathogen was present during root growth. Additionally, we also determined the expression of *PR1*, *POX*, and *PAL* in the seven selected cultivars. This in vitro cultivation method is similar to that introduced by Stephens and Elmer (1988) [8], who used it for evaluating *Asparagus* spp. against *F. oxysporum* and recommended it as a rapid resistance screening technique. Gossmann et al. (2011) [21] tested the pathogenicity of *Fusarium* spp. on young seedlings that grew in water agar in big test tubes filled with water agar. Our procedure differed from both in that we used only 5 μL of Foa1-spores suspension that was directly injected near the first root instead of floating 1 mL of Foa1-spores suspension on top of the medium. Through this minor inoculum quantity in the root zone, we wanted to ensure a systemic infection that would allow slow disease progression and the detection of a broader range of gene expression differences. At 5 dpi, the in vitro inoculated roots showed typical brownish lesions on the primary roots as well as on the lateral roots. Symptoms allocated to Fusarium infestation differed in length and shape and were clearly visible in the culture tubes of all Foa1-inoculated plants. However, when focusing on growth traits of the cultivars grown in vitro without the fungus, we could identify two distinguishable groups: one fast growing and one with slow root formation. Irrespective of inoculation, 'Vitalim', 'Gijnlim', and 'Backlim' belonged to the fast-growing group, and 'Grolim', 'Ramires', 'Rapsody', and 'Xenolim' belonged to the slow-growing group. Contrary to our expectations, Foa1 infestation did not impair primary root growth and the development of lateral roots; it even stimulated both, as also observed in the sand cultivation screening. This phenomenon was also observed by Jacobi et al. (2023) [11], who therefore recommend not to rely on the reduction in root growth as a criterion for evaluating susceptibility. However, in our view, this could also be understood as a self-defense strategy of affected plants trying to escape the pathogen by accelerating root growth and root formation. To obtain some insights into this self-defense strategy, we investigated specific expression differences in *PR1*, *POX*, and *PAL* as markers of infection progression and plant defense response. In contrast to the sand culture, a significantly strong induction of the expression of all three analyzed genes was found in the roots of all of the investigated cultivars, which was accompanied by clearly visible symptoms on all roots and therefore expected. Interestingly, 'Xenolim', the cultivar with the lowest level of symptomatic roots, showed the lowest induction of *PR1*. The PR1 protein is known to have direct antimicrobial activity against different plant pathogens; however, its putative function against Fusarium remains to be determined [22]. On the other hand, recent results have shown that an effector of *Fusarium oxysporum* f. sp. *lycopersici* binds tomato PR1 and translocates it into the host nucleus, ultimately facilitating the pathogen´s virulence [23]. Expression of the *POX* gene was most highly induced in the 'Vitalim' and 'Xenolim' roots inoculated with Foa1 in vitro. Since the *POX* gene product is involved in cell wall lignification [18], it could be assumed that the reduced symptomatology on the roots, especially in 'Xenolim', is due to increased lignification impeding penetration of the pathogen. A significant induction of *PAL* was also found in all roots inoculated with Foa1 in vitro. Here, the smallest increase in expression was detected in the cultivars 'Grolim', 'Ramires', 'Rapsody', and 'Xenolim'. Interestingly, these varieties showed significantly increased growth of the primary root at 5 dpi plus increased lateral root formation. Since *PAL* is involved in SA biosynthesis [19,20], one could assume that lower synthesis of SA would result in less growth inhibition compared to the defense response. However, it is questionable whether increased expression also translates into increased activity of the encoded proteins. A publication by He et al. (2001) [24] suggests that, in particular, in the susceptible *Asparagus officinalis*, no increased activity of *POX* and *PAL* can be measured after inoculation with *F. oxysporum* f. sp. *asparagi* or *F. proliferatum*, whereas in the resistant *Asparagus densiflorus* vars. Myersii and Sprengeri, a clear induction was measured after inoculation. Future experiments will show whether increased expression will also be followed by increased activity in the varieties used here.

In contrast, shoot development was slightly restrained by inoculation with the pathogen. At 5 dpi, the Foa1-inoculated plants developed fewer shoots, except for 'Grolim'. This result

matches with the gene expression results for shoots where the relative expression difference in *PR1* was highest in 'Grolim' yet not significantly different from the other cultivars.

Plant roots are known to secrete metabolites to attract, feed, or repel microorganisms [25]. The in vitro culture experiment confirmed that the longer the 1st primary root, the longer the downward growth of the Foa1 isolate. This may suggest that the fungus grew along the roots due to the exudates released by the roots [26]. Future experiments will show whether root exudates differ in their metabolomic signature and how Fusarium growth is dependent on the type of exudates. Furthermore, this in vitro method could be a new system to collect root exudates of asparagus cultivars and use them for the growth testing of *Fusarium* spp.

Whether the earlier induction of defense-related genes in 'Rapsody' and 'Xenolim' cultivated in sand provides the plant with better resilience to Fusarium needs to be investigated in further, longer-term experiments. Furthermore, it is of relevance whether the root exudates in sand cultivation differ significantly between the cultivars and thus have an influence on the growth and development of Fusarium or whether morphological differences of the roots cause different colonization of the plants.

The comparison of two resistance screening methods for evaluating asparagus resistance against *F. oxysporum* f. sp. *asparagi* suggests that although sand cultivation is indeed the more natural method for testing plants against pathogens, more time is needed for evaluating disease symptoms due to the long duration of host–pathogen establishment. The disadvantage here is that the longer cultivation takes, the more environmental influences alter the molecular responses of the plant. Pulling the roots out of the sand was still feasible about 20 days after sowing, but when the screening takes longer, pulling the uninjured roots out of the sand would most likely be hampered, negatively affecting the analyses of root morphology and molecular defense response. In contrast, the in vitro resistance test is a rapid test that allows screening of a large number of asparagus genotypes. Moreover, the fungal growth can be continuously and easily monitored together with the root traits with a ruler. Another advantage is that the roots do not need to be pulled out to measure the roots' growth and development. For performing molecular analyses and taking root images at the end of the experiment, it is very easy to remove the roots from the medium without injuring them. The drip inoculation of roots with a certain amount of spore suspension is a reliable approach for both methods. In contrast, dip-inoculation of the roots may not be uniform, since pulling out the roots may cause an uneven number of injuries, making the treatment with Foa1 spores irregular.

**Supplementary Materials:** The following supporting information can be downloaded at: https://www.mdpi.com/article/10.3390/horticulturae9060656/s1. Table S1: Fresh mass (FM) and standard deviation (SD) of roots and shoots of in sand cultivated asparagus cultivars; Table S2: Fresh mass (FM) and standard deviation (SD) of roots and shoots of in vitro cultivated asparagus cultivars.

**Author Contributions:** Conceptualization, R.D.F.-K. and R.Z.; disease assessment, R.D.F.-K.; molecular analysis, R.Z.; imaging analysis, J.G.; writing—original draft preparation, R.D.F.-K. and R.Z.; writing—review and editing, R.D.F.-K., J.G. and R.Z. All authors have read and agreed to the published version of the manuscript.

**Funding:** This research received no external funding.

**Data Availability Statement:** The data presented in this study are available from the corresponding author upon request.

**Acknowledgments:** We thank Hans Maximilian Blum (Technical University Berlin, Germany) for his valuable work on the data collection for the in vivo experiment and Katja Witzel (IGZ, Germany) for critically reading the manuscript.

**Conflicts of Interest:** The authors declare no conflict of interest.

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
