# Peer review of "Root Growth and Defense Response of Seedlings against Fusarium oxysporum in Sand Culture and In Vitro—A Comparison of Two Screening Approaches for Asparagus Cultivars"

_horticulturae, doi:10.3390/horticulturae9060656_

Round 1
Reviewer 1 Report
Two rapid asparagus screening methods, in sand culture and in vitro, were tested to evaluate response of young seedlings against Fusarium oxysporum f. sp. asparagi. This study has a certain degree of innovation. The research results provide new approachs for the breeding of asparagus, which has great application value. The comments on each part of the article are as follows:
1. Title
The title is reasonable and informative.
2. Abstract
The length of the abstract is reasonable and provides a good summary of the paper.
I suggest the author directly explain which method is more feasible according to their results.
3. Keywords
The keywords are reasonable and representative.
3. Introduction
The Introduction provides a good overview of the background of this study, with comprehensive references. Some suggestions of this part are listed below:
1) I suggest explaining the Latin name of asparagus in the first sentence, so that readers can understand it more easily.
2) Line 48: Fusarium, I think this is a latin genus' name, so it should be in italic form. Or, we can use "fusarium" instead of "Fusarium". So as Line 61.
3) Line 79: "of two test methods" better be "of the two tested methods"?
4. Materials and Methods
The method design is reasonable and the description is specific.
Suggestions:
1) Line 91: "Tween20" or "Tween 20"? With a space between as you did in line 90.
2) Line 97: I suggest listing the conidia concentrations in this text. As a reader, I would not like to read reference [12] to get this short but important information.
3) Line 112: including 4 plants each?
4) Line 121: 600 DPI, please Please provide the full expression of "DPI" as you have "dpi" through the context.
5) Line 191 and 121: Why the roots were scanned at different DPI?
6) Is it better to combine 2.4 and 2.6?
5. Results
The results are clearly presented. The data are presented in good form.
Suggestions:
1) Is it better to present the results in these parts: 3.1 morphological assessment, 3.2 gene expression. I think that this writing structure is better for comparing the results of the two methods.
2) I suggest providing pictures of the symptoms.
3) Should the bar charts in the text have error bars?
6. Discussion
The discussion was very in-depth and compared well with similar studies.
7. References
1) Line 517-519: Do the first letters of the article title need to be capitalized?
2) Fusarium oxysporum, italic. So as the other Latin species names of the references.
Minor editing of English language required
Author Response
Thank you very much, and please see the attachment.

Reviewer 2 Report
Dear Authors,
The methodology is presented in detail and consistently. The experimental design is appropriate and the manuscript is well-structured. Results and discussion described reliably and clearly.
Author Response

(The authors gave the same response as above.)
